# Impact of online lottery sales prohibition on the structure of lottery consumers: A time cost perspective in China

Zeyu Feng ⓘ*

School of Mathematics, Southwestern University of Finance and Economics, Chengdu, Sichuan, China

* sxjl3496@vip.qq.com

## Abstract

Lottery public welfare funds constitute a significant source of government fiscal revenue. Consequently, it is of crucial importance to objectively analyze the impact of purchasing convenience on the lottery industry. This study employs the discrete-time dynamic optimization Bellman equation to examine the influence of time costs associated with purchasing lottery tickets on consumer demand across different income groups, considering various types of lottery tickets. Utilizing panel data spanning from 2007 to 2021 from 30 provinces in China, the Difference-in-Differences (DID) method is employed to assess the effects of the ban on online lottery sales, implemented in 2015, on the sales of welfare and sports lottery tickets in distinct income regions. The findings indicate that the prohibition of online lottery sales leads to a more rapid decline in the demand for betting-lottery tickets in high-income areas compared to low-income areas, while no significant variation in the rate of decline in demand for lotto-lottery tickets is observed. This trend hampers the improvement of the lottery consumer structure and overall sales. The research outcomes furnish valuable references for enhancing the structure of lottery consumers and augmenting sales within the lottery industry.

## Introduction

Globally, wealth inequality remains a central challenge in social development. As a unique public fundraising mechanism, lotteries have drawn significant academic interest regarding their impact on wealth redistribution (Yao et al., 2024 [1]). While lottery revenues are explicitly allocated to social welfare programs in many nations (Mikesell et al., 1994 [2]; Walker, 2003 [3]; Hansen, 2008 [4]), empirical evidence reveals a structural imbalance in purchasing demographics. Low-income groups disproportionately dominate lottery consumption (Garrett, 2001 [5]), undermining the redistributive equity of lottery systems and potentially exacerbating *"reverse redistribution"* effects—where socioeconomically vulnerable populations become primary contributors to public funds. This paradox motivates our central research inquiry: How can

**Data availability statement:** All Dataset and STATA DO files are available from figshare (https://doi.org/10.6084/m9.figshare.30304450.v2) or protocols.io (DOI: dx.doi.org/10.17504/protocols.io.n92ld6qe8g5b/v1 (Private link for reviewers: https://www.protocols.io/private/0AED7227A2A211F0B9FF0A58A9FEAC02 to be removed before publication.)) or the OPENICPSR database (URLs: https://www.openicpsr.org/openicpsr/project/236845/version/V1/view Project Citation: Feng, Zeyu. Impact of Online Lottery Sales Prohibition on the Structure of Lottery Consumers: A Time Cost Perspective in China. Ann Arbor, MI: Inter-university Consortium for Political and Social Research [distributor], 2025-07-23. https://doi.org/10.3886/E236845V1).

**Funding:** The author(s) received no specific funding for this work.

**Competing interests:** The authors declare that they have no conflict of interest.

influencing factors of lottery purchasing behavior (particularly the time cost) be leveraged to optimize purchaser demographics by increasing high-income groups' participation, thereby enhancing the equity and societal efficacy of lottery-based welfare systems?

China's lottery market—operating under a state monopoly administered exclusively by the China Sports Lottery (CSL) and China Welfare Lottery (CWL)—serves as an ideal case study. The CSL offers numerical lotteries such as Super Lotto, sports-betting games such as football match result lottery, and instant scratch cards, while the CWL focuses on traditional draw games such as Double Color Ball, alongside instant products. For distribution, over 95% of transactions have relied on over 200,000 physical outlets nationwide since the 2015 online sales ban; consumers pay only the face value of tickets (such as ¥2 per bet for Double Color Ball), with no brokerage fees.

According to La Fleur's 2024 World Lottery Almanac, China achieved record-breaking lottery sales of $418 billion in 2023(Converted at the 2023 average exchange rate of 7.10 RMB/USD; the same rate applies throughout this analysis) . Despite leading global sales volume, China's annual per capita consumption expenditure on lottery products ($49) exhibits marked regional heterogeneity. This figure significantly trails that of high-income nations such as Norway ($864) and Switzerland ($632) in terms of annual per capita lottery expenditure. and even trailing some middle-income economies. The disparity between aggregate sales and per capita consumption suggests significant variations in income-group participation rates. Significantly, after China imposed the internet lottery sales ban in 2015 (substantially reducing accessibility), the annual growth rates of China Welfare Lottery and China Sports Lottery plummeted from pre-ban levels of 16% and 30% to -7% and -18% respectively, empirically substantiating the critical role of purchasing convenience in shaping lottery consumption patterns.

This study focuses on the pivotal behavioral variable of "time cost," integrating theoretical analysis and empirical approaches to address three sequential research questions: (a) Does the reduction in lottery purchasing accessibility disproportionately affect high-income or low-income groups? (b) Are there systematic differences in such impacts across lottery product categories (lotto, sports-betting)? (c) Can strategic time-cost optimization be employed to adjust lottery participant demographics?

Theoretically, we construct a dynamic optimization model to dissect the intrinsic mechanisms through which time costs influence consumption decisions. Empirically, utilizing the exogenous policy shock of China's 2015 online lottery sales ban, we implement a Difference-in-Differences (DID) framework to identify heterogeneous effects of accessibility changes across regions with varying income levels. Results demonstrate that increased time costs cause demand for sports-betting lotteries to decline significantly faster among high-income populations than low-income groups, a pattern absent in other lottery types. This finding not only provides a novel theoretical explanation for "low-income lottery consumption preferences" but also offers critical policy implications: differentiated time-cost thresholds across lottery varieties could enable precise adjustment of participant demographics.

Broadly, this study informs the optimization of China's "third distribution" mechanism. Against the national strategic backdrop of common prosperity, our research validates behavioral economics tools' unique capacity to enhance equity in public fundraising systems, providing scientific evidence for constructing more equitable and sustainable social welfare financing architectures.

## Review

Modern-style lotteries emerged as early as the 15th century in Europe and became more prevalent by the 18th century, with early scholars having explored lottery-related topics across various fields. However, systematic academic research on lottery participation behavior within the modern economic framework is notably rooted in the expected utility theory proposed by Von Neumann and Mogenstern (1944) [6], which provided core analytical tools for subsequent studies. This theory was subsequently challenged by Friedman and Savage (1948) [7], who observed that risk-seeking individuals purchasing lotteries simultaneously acquire insurance products, exhibiting risk aversion. This coexistence of risk aversion and risk-seeking behavior, termed the "Friedman–Savage Puzzle," led to their proposal of a utility function combining quasi-concave and convex segments to explain this phenomenon. Subsequently, Savage (1954) [8] extended this framework by introducing subjective probabilities into utility functions, establishing the subjective expected utility theory.

These theories were subsequently challenged by the Allais Paradox (Allais, 1953) [9], the Ellsberg Paradox (Ellsberg, 1961) [10], and critical refutations such as Bailey et al. (1980) [11]. While later theories advanced frameworks for addressing uncertainty (Anscombe and Aumann, 1963 [12]; Rothschild and Stigliz, 1970 [13]; Lichtenstein and Slovic, 1971 [14]), they failed to fully resolve the two paradoxes.

The resolution of the Allais Paradox became feasible only with Kahneman and Tversky's (1979) [15] introduction of prospect theory. For addressing the Ellsberg Paradox, the most influential contribution emerged from Gilboa and Schmeidler's (1989) [16] Maxmin Expected Utility (MEU) theory. Subsequently, Gilboa and Schmeidler (1995) [17] employed a babysitter hiring example to illustrate decision-makers' propensity to select actions maximizing similarity-weighted expected value. However, both prospect theory and MEU theory have limitations in explaining real-world lottery behavior: prospect theory, focused on static decisions and hindered by individual parameter variability, fails to account for dynamic adjustments like loss chasing and social influences, while MEU's assumptions of perfect rationality and fixed risk attitudes cannot capture overestimation of small probabilities, entertainment motives, or context-dependent risk preferences—limitations amplified by digital lotteries, where gamified features like instant feedback distort risk perceptions beyond these models' scope.

A seminal contribution by Conlisk (1993) [18] posits that utility functions can rationalize the "Friedman-Savage Puzzle" provided lotteries generate supplementary utility, termed non-wealth motivations such as entertainment, social interaction, or the psychological thrill of anticipation. Extensive empirical investigations confirm that such non-monetary benefits systematically influence lottery purchase decisions (Sauer, 1998 [19]; Forrest et al., 2002 [20]; Humphrey, 2013 [21]; Mathieu et al., 2020 [22]).

Thaler's (1985) [23] seminal work revealed that consumers mentally frame lottery purchases as expenditures from entertainment mental accounts, further identifying the House Money Effect (Thaler, 1990 [24]). Nyman et al. (2008) [25] identified that lottery demand originates from a "something for nothing" motivation, contending that lottery gains constitute not merely supplemental income but labor-cost-free windfalls, with explicit incorporation of labor supply into their analytical framework. Stetzka (2023) [26] synthesizes lottery behaviors as hybrid manifestations of Full Rationality, Bounded Rationality, and Irrationality: majority participation driven by consumption motives with cognitive bias distortions, juxtaposed against a minority exhibiting irrational problem gambling. Cameron's (2024) [27] utility-theoretic model decouples gambling behaviors into two distinct phases by integrating "Stock of Past Gambling", resolving the paradox of deliberate engagement in knowingly detrimental activities.

Lottery participation emerges from multifaceted motivations: megalithic jackpot fantasies, non-wealth incentives (e.g., entertainment/socialization), and interactive tensions between cognitive biases and imperfect rationality.

Scholars have further examined lottery participation through socioeconomic lenses including educational attainment, age, gender, and familial gambling culture. Consistent evidence indicates heightened propensity among populations with low educational attainment, advanced age, and family traditions of lottery engagement (Elisabeth et al., 2015 [28]; Brochado, 2018 [29]; lutter et al., 2018 [30]; Chagas et al., 2022 [31]). Gabrielyan and just(2020) [32] demonstrated unemployment rate escalation correlates with increased traditional lottery sales while exerting negligible impact on instant games, underscoring the necessity of product-type differentiation. Fu (2021) [33] identified pronounced fixed-prize lottery participation in low-income communities, followed by instant games, with minimal participation disparities in progressive jackpot formats.

A critical examination of demographic literature reveals lottery participation as a multiplicity of interacting determinants, spanning individual traits (income level, age, education, gender), environmental factors (community norms, advertising intensity, accessibility of outlets), and game mechanics (jackpot magnitude, product design).

Within the research domain of temporal costs' influence on consumption demand, Becker (1965) [34] conceptualized the economic valuation of time expenditure in purchasing behavior. Empirical consensus indicates that elevated temporal costs generally depress consumer satisfaction and demand for goods while inducing deferred purchases of low-priced commodities (Dhar and Nowlis ,1999 [35];Chernev's study ,2003 [36];Nowlis et al. ,2004 [37];Botti and Iyengar's research,2006 [38]).

The proposition that online shopping amplifies product demand through time cost reduction is empirically corroborated by cumulative research findings (Lennon et al. (2008) [39]; Mittal (2013) [40]; Akroush and Al-Debei (2015) [41]; Huseynov et al. (2016) [42]; Saha et al. (2020) [43]). Extending this logic to the gambling sector, early explorations of mobile gambling highlighted distinct psychological and structural mechanisms reshaping behavior: digital platforms, enabled by smartphone accessibility, interact with associative learning processes to accelerate maladaptive gambling behaviors, while digitized lottery markets disproportionately drive addiction due to omnilocal accessibility and transactional immediacy—factors that eliminate physical constraints, truncate rational deliberation intervals, and amplify impulsive decision-making (James et al. (2017) [44]; Churchill and Farrell (2018) [45]; Zhang Chen et al., 2022 [46]).

Subsequent investigations into online/mobile gambling engagement, particularly within sports betting, further clarified key drivers: real-time excitement, smartphone-enabled accessibility, and features like "cash-out" enhance perceived control over outcomes; promotional offers and participation excitement shape young adults' habits, with this demographic often underestimating marketing risks to vulnerable groups; targeted strategies (e.g., enhanced odds) reduce perceived risk and fuel impulsive betting by fostering illusions of control; and online sports gambling has become normalized among college students as an extension of gaming, with states increasingly adopting it for revenue despite associated harms (Killick and Griffiths (2021) [47]; Dunlop and Ballantyne (2021) [48]; Killick and Griffiths (2022) [49]; Reeve and Pincin (2025) [50]).

Recent research underscored how digital efficiency drives escalated gambling expenditure: instantaneous, ubiquitous access to betting apps—coupled with platform functionality, social influences, and targeted marketing—facilitates harmful behaviors among young adults (e.g., increased frequency, impulsive wagering, and loss chasing); concurrently, U.S. states with high online lottery penetration exhibited faster post-pandemic revenue recovery, confirming that temporal and spatial efficiency acts as a critical driver of amplified lottery spending (Hing et al. (2024) [51]; Hickman (2025) [52]). Collectively, these studies highlight digitalization's multifaceted impact on gambling behavior, spanning psychological mechanisms, marketing dynamics, and demographic vulnerabilities

Existing research has primarily focused on three areas: lottery purchasing behaviors among low-income groups, macroeconomic analyses of lottery consumption, and the influence of temporal costs on consumer demand. However, within the context of digitalized gambling, these studies exhibit notable limitations. First, although some research has examined the impact of purchasing convenience on consumption, few have analyzed lottery consumer characteristics

from the perspective of purchasing convenience, with a particular lack of exploration into differences across income groups and lottery categories in this dimension. Second, traditional theories are mostly constructed based on offline scenarios and struggle to explain how features of the digital environment, such as real-time accessibility and mobile interactions, reshape lottery consumption decisions. Third, existing research on online gambling has largely concentrated on sports betting, with insufficient attention to the unique consumption mechanisms of online lotteries. Notably, lotteries and sports betting differ significantly in terms of participation motives (e.g., luck-oriented vs. skill-oriented) and audience structures.

These gaps highlight the value of this study: Based on the utility model framework (Ghez and Becker, 1975 [53]), we introduce a temporal cost perspective, incorporating leisure time into an intertemporal substitution analysis of lottery consumption. Leveraging China's unique policy context of "prohibiting online lottery sales" as a natural instrument, this study empirically explores the heterogeneous effects of temporal costs across income groups and lottery categories, thereby offering new insights into the mechanisms of lottery consumption.

## Theoretical model

Drawing insights from Conlisk's canonical model, this paper incorporates non-pecuniary utility into the consumption effect while accounting for time costs, with leisure likewise contributing to utility. To maintain model simplicity, we exclude non-wealth monetary utility, such as transactional convenience or subjective satisfaction from holding cash, while retaining the wealth utility of money, which remains central to the analysis. Within this adapted Money-in-Utility (MIU) framework, consumers optimize their time allocation between consumption and leisure, subject to the budget constraint specified below. The aggregate utility function takes the following form:

$$Max \sum_{t=0}^{\infty} \beta^t U(C_t, R_t) \tag{1}$$

where $0 < \beta < 1$ is a subjective rate of discount. $C_t$ is time $t$ consumption which include the consumption of lotteries and $R_t$ is the leisure enjoyed by consumers in time $t$. Utility is assumed to be increasing in both arguments, strictly concave and continuously differentiable. The demand for leisure will always be positive if we assume that $\lim_{R \to 0} U_R(C, R) = 0$ for all $C$, where $U_R = \frac{\partial U(C,R)}{\partial R}$.

### Model assumptions

In this model, it is important to establish definitions for the consumer's choices regarding consumption, time, leisure, labor, wage and wealth.

**Assumption for consumption.** The consumer's consumption in period $t$, $C_t$, is defined as follows:

$$C_t = C_t^o + T_t = C_t^o + T_t^b + T_t^l$$

Where $T_t$ represents the total amount spent on lottery purchases in period $t$; $T_t^b$ represents the consumer's expenditure on betting-type lottery ("Betting-type" is one of the major categories of lottery, with sports lottery being a prominent example. Over the past three years, the share of betting-type lottery has exceeded 50% of the total sports lottery in China.) in period $t$, $T_t^b \geq 0$; $T_t^l$ represents the consumer's expenditure on lotto-type lottery ("Lotto-type" is one of the primary categories of welfare lottery, representing a significant portion of the overall welfare lottery market. Its share within the welfare lottery sector has surpassed 75% in the past years.) in period $t$, $T_t^l \geq 0$; and $C_t^o$ denotes the aggregate consumption of other goods and services by the consumer in period $t$, with the condition $C_t^o > 0$ indicating the minimum level of consumption required for basic subsistence.

**Assumption for time.** Consumers' time endowment in each period is defined as a fixed constant, and the time endowment is normalized to "1" (excluding the time consumers must allocate for necessary rest). This period can be a day, a month, or even a year. Consumers' time endowment can be allocated to leisure (which brings utility to consumers), labor (to obtain labor income), and the time cost of purchasing lottery tickets (the time spent on purchasing lottery tickets).

The leisure enjoyed by the consumer in period $t$, $R_t$, is defined as:

$$R_t = R_t^o + R_t^b$$

In period $t$, $R_t^o$ represents other leisure time enjoyed , with the condition $R_t^o > 0$ indicating the consumer's inherent demand for leisure on a daily basis.

$R_t^b$ represents the time spent watching sports events in period $t$, which includes the time spent on watching sports news, information, research, and discussion, where $R_t^b \geq 0$.

It is also assumed that consumer spending on sports lottery tickets is a function of the time spent watching sports events, i.e., $T_t^b(R^b t)$, and that as the time spent watching sports events increases, consumer interest and utility in purchasing sports lottery tickets will increase , i.e., $\frac{\partial T_t^b}{\partial R^b t} > 0$. (Sports viewing activates "fast thinking," lowering rational evaluation thresholds and facilitating emotionally driven lottery purchases (Kahneman, 2011 [54]). Empirically, a significant positive correlation exists: during events like the World Cup, increased viewing duration directly stimulates purchases, and spectator engagement such as emotional investment enhances motives to drive lottery decisions (Lin et al., 2022 [55]; Yao et al., 2024 [56]))

The time cost of purchasing lottery tickets for consumers in period $t$ is defined as $n_t$. And we assume that $n_t$ only crowds out leisure time $R_t$.

When it is possible to purchase lottery tickets online, $n_t$ is very small and can even be considered equal to 0.

The labor time (or working time) spent by consumers in period $t$, $L_t$, and the time constraint is defined as follows:

$$R_t = R_t^b + R_t^o = 1 - L_t - n_t \tag{2}$$

where $L_t \geq 0$, indicates that the consumer has the option to choose not to work.

The above equation represents the time constraint. Here, 1 represents the standardized time endowment that consumers possess after deducting necessary rest time. This can be one day, one month, or even one year. The time endowment can be considered a fixed constant.

**Assumption for wealth.** Firstly, for the sake of analytical simplicity in subsequent analysis, the consumer's wage function $Y_t$ is defined as a linear function of labor supply $L_t$:

$$Y_t = Y(L_t, S_t) = b + S_t L_t \tag{3}$$

Here, $b$ represents the basic wage, $S_t L_t$ represents the performance wage, and $S_t$ represents the wage rate. $Y(L_t, S_t)$ is the wage function, which indicates that the income in period $t$ is positively related to the labor supply and the wage rate in period $t$, i.e., $\frac{\partial Y}{\partial L} > 0, \frac{\partial Y}{\partial S} > 0$.

Secondly, The consumer's wealth in period $t$, $W_t$, is defined as:

$$W_t = Y_t(L_t, S_t) + \frac{1 + i_{t-1}}{1 + \pi_t} A_{t-1} + J_t = C_t + A_t \tag{4}$$

Here, $A_{t-1}$ represents the financial assets held by the consumer in period $t$-1, including cash, large certificates of deposit, real estate, bonds, funds, and stocks, etc. $i_{t-1}$ represents the rate of return on financial assets in period $t$-1. $J_t$ represents the lottery winnings obtained by the consumer in period $t$.

Lastly, the consumer's expected wealth in period $t$+1, $W_{t+1}^E$, is defined as:

$$W_{t+1}^E = Y_{t+1}(L_{t+1}, S_{t+1}) + \frac{1 + i_t^E}{1 + \pi_{t+1}} A_t + J_{t+1}^E(T_t)$$

Here, the superscript $E$ represents the consumer's subjective expectations about the future. $i_t^E$ represents the consumer's subjective expected rate of return on financial assets in period $t$, and $J_{t+1}^E(T_t)$ represents the consumer's subjective expected returns on lottery tickets in period $t$+1. It is assumed that $\frac{\partial J_{t+1}^E(T_t)}{\partial T_t} > 0$ and is denoted as $J_{T_{t+1}}^E = \frac{\partial J_{t+1}^E(T_t)}{\partial T_t}$. For the sake of analytical simplicity in subsequent analysis, It is assumed that the inflation rate in the current period is the same as the inflation rate in the subsequent period, i.e., $\pi = \pi_t = \pi_{t+1}$, and that the consumer has the same subjective expectations about the future in each period, so the consumer's subjective expected rate of return on financial assets and the marginal returns on lottery tickets are assumed to be the same in the current period and the subsequent period. i.e., $i^E = i_t^E = i_{t+1}^E; J_T^E = J_{T_{t+1}}^E$. It is also assumed that the marginal returns on lottery tickets are constant over time, and therefore, the expected lottery prize function can be expressed as a linear function of lottery ticket consumption:

$$J_{t+1}^E(T_t) = P^E K T_t = \sigma^E T_t$$

Here, the consumer's subjective expected probability of winning the lottery, $P^E$, and the subjective expected marginal returns on lottery tickets, $\sigma^E$, are the same in each period and do not vary over time. $K$ represents the lottery jackpot, which is a fixed constant. With these assumptions, The transition equations for $W_t$ and $W_{t+1}^E$ can be expressed as:

$$W_t = Y_t(L_t, S_t) + \frac{1 + i_{t-1}}{1 + \pi_t}(W_{t-1} - C_{t-1}) + J_t$$

$$W_{t+1}^E = Y_{t+1}(L_{t+1} S_{t+1}) + \frac{1 + i^E}{1 + \pi}(W_t - C_t) + J_{t+1}^E(T_t) \qquad (5)$$

Since it is assumed that $i^E$ and $\pi$ are constant over time, it is possible to simplify the equation by defining $I^E = \frac{1+i^E}{1+\pi}$. $I^E$ is a constant that represents the real expected rate of return on a series of financial assets after adjusting for inflation.

## Demand functions

In this section, our objective is to analyze the effects of the ban on online lottery sales on the demand for lotto lottery and betting lottery, and its implications for the income structure of lottery consumers. To achieve this, we will derive the demand functions of consumers for these two types of lotteries, taking into consideration the increased time cost associated with purchasing lottery tickets. By examining the changes in consumer behavior and income distribution, we can gain insights into the overall impact of the policy on the lottery market.

**The value function.** The consumer's problem is to choose paths for $C_t$, $R_t$, and $A_t$ to maximize the total utility (1) subject to the wealth constraint (4) and leisure time (2) constraint. This is a problem in discrete-time dynamic optimization. To find the optimal solutions, we employ the Bellman equation (Bellman, 1957) (We provide references to the seminal works on the derivation and proof of the Bellman equation (Bellman, 1957) [57], Lucas (1978) [58] and Benveniste and Scheinkman (1979) [59].) , a key tool in discrete-time dynamic optimization. The value function gives the maximized value

of utility the consumer can achieve by behaving optimally, given its current state. (For introductions to dynamic optimization designed for economists see Sargent (1987) [60], Lucas and Stokey (1989) [61], Dixit (1990) [62], Chiang (1992) [63], Obstfeld and Rogoff (1996) [64], Ljungquist and Sargent (2000) [65] or Walsh(2010) [66].) The state variable for the problem is the consumer's initial resources $W_t$. The value function, defined as the present discounted value of utility if the consumer optimally chooses consumption, leisure and financial asse, is given by:

$$V(W_t) = max[U(C_t, R_t) + \beta V(W^E_{t+1})] \qquad (6)$$

Here, $V(W_t)$ represents the value of wealth in period $t$, $U(C_t, R_t)$ is the utility function determined by consumption $C_t$ and leisure $R_t$ in the current period, and $\beta V(W^E_{t+1})$ is the discounted value of future wealth, which is also the capital to enjoy consumption in the future.

The maximization is subjecting to the wealth constraint (4) (5) and leisure time inequality constraint (2), we have:

$$W_t = Y_t(L_t, S_t) + \frac{1 + i_{t-1}}{1 + \pi_t} A_{t-1} + J_t = C_t + A_t$$

$$W^E_{t+1} = Y_{t+1}(L_{t+1}, S_{t+1}) + \frac{1 + i^E}{1 + \pi}(W_t - C_t) + J^E_{t+1}(T_t)$$

$$R^o_t + R^b_t = 1 - L_t - n_t \geq 0$$

By applying the first-order conditions of decision variables and employing the envelope theorem with respect to state variables(detailed derivation process can be found in S1 Appendix), we can obtain the Euler equation for lottery consumption:

$$\frac{U_{T^b_t}}{U_{T^b_{t+1}}} = \frac{U_{T^l_t}}{U_{T^l_{t+1}}} = \beta I^E \qquad (7)$$

The economic implication of the above equation is that the greater the subjective expected return on financial assets $I^E$ for the consumer, the greater the marginal utility of lottery consumption $U_{T_t}$ in the current period, and the smaller the quantity of lottery consumption $T_t$ in the current period; or the smaller the marginal utility of lottery consumption $U_{T_{t+1}}$ in the future, and the greater the quantity of lottery consumption $T_{t+1}$ in the future. Similarly, any information that lowers the consumer's subjective expected return on financial assets (such as stock market crashes, economic recessions, or severe inflation) or personal inability to obtain returns from financial assets will increase the consumer's demand for lottery consumption in the current period.

In order to analyze the effect of purchasing time on consumer's lottery consumption, it is necessary to obtain the relationship between leisure and lottery demand(detailed derivation process can be found in APPENDIX A). We can derive the following relationship:

$$U_{T^b_t}\left(\frac{I^E \frac{\partial Y_t}{\partial L_t}}{I^E - \sigma^E_s}\right) = U_{T^l_t}\left(\frac{I^E \frac{\partial Y_t}{\partial L_t}}{I^E - \sigma^E_w}\right) = U_{R^o_t} \qquad (8)$$

In the above equation, since the marginal utility of lottery and leisure cannot be negative, it must hold that $I^E - \sigma^E_w > 0$ (If $I^E < \sigma^E_w$ consumers would abandon a series of financial assets (including cash, deposits, real estate, funds, stocks, bonds, etc.) and use them all to buy lottery, which is very unrealistic.) ,which means that the subjective marginal return of lottery cannot be higher than that of financial assets. The marginal effect of consuming lottery is positively related to the marginal utility of leisure. Therefore, when the purchase time increases and the marginal utility of leisure increases, the marginal

effect of lottery also increases accordingly, resulting in a decrease in lottery consumption. For the sake of simplicity in analysis, the next section will separately consider the consumption of betting-lottery and lotto-lottery.

**CRRA utility function.** To facilitate the examination of the influence of time cost on lottery sales, this section will introduce the constant relative risk aversion (CRRA) utility function. The CRRA utility function is particularly suitable for analyzing intertemporal issues and aligns with the approach established by Merton (1971) [67] [68]:

$$U(C) = \frac{C^{1-\theta} - 1}{1 - \theta}$$

The parameter $\theta$ is known as the coefficient of relative risk aversion ($\theta$ is the elasticity of marginal utility with respect to consumption, i.e., $\theta(C_t) = -\frac{\partial(\ln U'(C_t))}{\partial(\ln(C_t))} = -\frac{C_t U''(C_t)}{U'(C_t)}$.), and the larger the value of $\theta$, the more risk-averse the consumer is towards future uncertainty. Let $\rho = \frac{1}{\theta}$, which is known as the intertemporal elasticity of substitution ($\rho$ is the intertemporal substitution elasticity, which reflects the relationship between the rate of change in consumption and the rate of change in marginal utility: $\rho = -\frac{\partial(\dot{C}_t/C_t)}{\partial(\dot{U}'(C_t)/U'(C_t))}$.). The value of $\rho$ determines how easily current consumption can be replaced by future consumption, with larger values of $\rho$ indicating easier substitution between present and future consumption and smaller values indicating a greater difficulty in substitution.

**CRRA Utility Function for Lotto-lottery.** In this section, we will analyze in detail the impact of high wages and high income on the sales of lotto and betting lottery tickets, as well as the effect of the intermediate time cost of purchasing lottery tickets on different income groups. Firstly, we will analyze the factors that affect the sales of welfare lottery tickets and how the purchasing time cost affects their sales. To simplify the analysis, we introduce the CRRA utility function for Lotto-lottery tickets. Since Lotto-lottery tickets are the same for consumers in each period and the probability of winning is very low, consumers generally do not consume them in large quantities at once, but distribute their consumption evenly over different periods. From the monthly sales data of Lotto-lottery tickets, it can be seen that the demand for them does not fluctuate significantly. Therefore, we assume that consumers rarely engage in intertemporal substitution for Lotto-lottery tickets, that is, $\theta^I > 1, \rho^I < 1$, and the CRRA utility function for Lotto-lottery tickets is expressed as follows:

$$U(T_t^I) = \frac{(T_t^I)^{1-\theta^I} - 1}{1 - \theta^I} \tag{9}$$

Analogously, after accounting for work hours, consumers generally enjoy a certain level of leisure in each period, avoiding extensive intertemporal substitution. For the sake of analytical simplicity, we assume that consumers exhibit the same intertemporal substitution behavior for leisure as they do for lottery consumption, i.e., $\theta^o = \theta^I > 1, \rho^o = \rho^I < 1$. The leisure CRRA (Constant Relative Risk Aversion) utility function can be expressed as follows:

$$U(R_t^o) = \frac{(R_t^o)^{1-\theta^I} - 1}{1 - \theta^I} \tag{10}$$

By incorporating equations (3), (9), and (10) into equation (8), we derive the following result:

$$T_t^I = \left(\frac{I^E S_t}{I^E - \sigma_w^E}\right)^{\rho^I} R_t^o \tag{11}$$

From the aforementioned equation, it is evident that the demand for lottery consumption is positively associated with the demand for leisure. When leisure $R_t^o$ increases by one unit, the demand for lottery consumption $T_t^I$ rises by $\left(\frac{I^E S_t}{I^E - \sigma_w^E}\right)^{\rho^I}$ units, implying that unemployed individuals may purchase a considerable amount of lottery tickets. Based on the definition in equation (2), if the lottery purchase time cost $n_t$ increases, leisure $R_t^o$ decreases, resulting in a drop in the demand

for lottery consumption $T_t^l$. Furthermore, it can be deduced from equation (11) that a rise in wages might also reduce consumer demand for lottery consumption; however, due to the intertemporal substitution elasticity $\rho^l < 1$, the impact of wage increases may not be significant, particularly when $\lim \rho \to 0$, as wage variations have little to no effect on lottery demand. Likewise, when the purchasing time cost $n_t$ increases, provided that $\rho^l$ is relatively small, there will be no significant disparity in the rate of decline in lottery consumption demand $(\frac{I^E S_t}{I^E - \sigma_w^E})^{\rho^l}$ between high-wage and low-wage individuals.

**CRRA Utility Function for Betting-lottery.** In this section, we examine the second scenario, exploring the relationship between wages and lottery purchase time costs and their influence on lottery sales. To simplify the analysis, we assume that consumers exclusively engage in sports lottery betting-type tickets and incorporate a CRRA utility function for sports lottery betting-type tickets. As the events associated with each period of betting-type lottery tickets differ, consumers are more inclined to consume larger quantities of sports lotteries during periods featuring captivating matches, such as the World Cup, European Cup, or various championship finals. This suggests that the cross-period demand elasticity for sports lottery consumption is relatively high, i.e., $\theta^b < 1$, $\rho^b > 1$. Consequently, the CRRA utility function for sports lottery betting-type tickets can be represented as follows:

$$U(T_t^b) = \frac{(T_t^b)^{1-\theta^b} - 1}{1 - \theta^b} \tag{12}$$

Likewise, each period's sports events differ in terms of their nature and levels of excitement, resulting in a current utility of watching sports events analogous to that of sports lottery betting-type tickets. Consumers tend to watch more sports events during periods featuring betting events. For the sake of analytical simplicity, we similarly assume that consumers demonstrate a relatively high cross-period demand elasticity for the duration of watching sports events $R_T^s$, i.e., $\theta^b < 1$, $\rho^b > 1$. The CRRA utility function for leisure spent watching sports events can be represented as:

$$U(R_t^b) = \frac{(R_t^b)^{1-\theta^b} - 1}{1 - \theta^b} \tag{13}$$

By incorporating equations (24), (3), (12), and (13) into equation (8), we derive the following result:

$$T_t^b = (\frac{I^E S_t}{I^E - \sigma_s^E})^{\rho^b} R_t^b \tag{14}$$

As evidenced by the aforementioned equation, an increase in the time consumers devote to watching sports events $R_t^b$ leads to a rise in the demand for betting-type lottery tickets $T_t^b$. In accordance with the definition in equation (2), if the lottery purchase time cost $n_t$ escalates, the duration of watching sports events $R_t^b$ declines, resulting in a drop in the demand for betting-type lottery tickets $T_t^b$. Given that $\rho^b > 1$, the rate of decline $(\frac{IS_t}{I^E - \sigma_s^E})^{\rho^b}$ is contingent on the magnitude of the wage $S_t$. A higher wage $S_t$ corresponds to a more rapid decrease in consumer demand for betting-type lottery tickets $T_t^b$ when the purchasing time cost $n_t$ increases. Consequently, high-wage individuals experience a more pronounced decline in demand for betting-type lottery tickets compared to low-wage individuals when faced with an increased lottery purchase time cost, leading to a substantial disparity.

### Analytical interpretation

In this section, we will utilize the derived consumer demand expressions for lotto-lottery and betting-lottery from the previous section to analyze how the increase in time cost due to the prohibition of online lottery sales affects the income structure of lotto-lottery and betting-lottery participants. We divide the consumer demand expressions (11) and (14) obtained

earlier based on consumer income into two categories: high-income and low-income. Consequently, we obtain the following four equation expressions:

$$Low-lotto: T_t^{LI} = (\frac{S_t^L}{1-\frac{\sigma_s^E}{I^E}})^{\rho^I} R_t^o$$

$$High-lotto: T_t^{HI} = (\frac{S_t^H}{1-\frac{\sigma_s^E}{I^E}})^{\rho^I} R_t^o$$

$$Low-betting: T_t^{Lb} = (\frac{S_t^L}{1-\frac{\sigma_s^E}{I^E}})^{\rho^b} R_t^b$$

$$High-betting: T_t^{Hb} = (\frac{S_t^H}{1-\frac{\sigma_s^E}{I^E}})^{\rho^b} R_t^b$$

In the equations above, "Low-lotto" refers to the demand of low-income participants for lotto-lottery, denoted as $T^{LI}t$ (similar notation applies to the following equations). "High-lotto" represents the demand of high-income participants for lotto-lottery, denoted as $T^{HI}t$. "Low-betting" denotes the demand of low-income participants for betting-lottery, expressed as $T^{Lb}t$. "High-betting" signifies the demand of high-income participants for betting-lottery, expressed as $T^{Hb}t$.

From these four equations, it is evident that regardless of the income level, when online lottery sales are prohibited, the increase in $n^t$ will displace $R_t^o$ or $R_t^b$, thereby reducing the current lottery demand for participants. Furthermore, the extent of demand reduction for both types of lotteries depends on the level of wages $S_t$ and the magnitude of $\rho$. Since lotto-lottery exhibits low intertemporal substitution elasticity (assuming $\rho^I$ is nearly 0), the slopes of $T^{LI}t$ and $T^{HI}t$, namely, $(\frac{S_t^L}{1-\frac{\sigma_s^E}{I^E}})^{\rho^I}$ and $(\frac{S_t^H}{1-\frac{\sigma_s^E}{I^E}})^{\rho^I}$, are both close to 1, indicating minimal differences. However, betting-lottery shows high intertemporal substitution elasticity (assuming $\rho^b > 1$), resulting in significant variations in the slopes of $T^{Lb}t$ and $T^{Hb}t$, namely, $(\frac{S_t^L}{1-\frac{\sigma_s^E}{I^E}})^{\rho^b}$ and $(\frac{S_t^H}{1-\frac{\sigma_s^E}{I^E}})^{\rho^b}$, due to different wages $S_t$ of participants. To facilitate a more intuitive analysis, we present these four equations graphically as shown below (Fig 1):

In the figure, the horizontal axis represents the leisure time enjoyed by lottery players, and the vertical axis represents their demand for lottery tickets. $R_t$ denotes the leisure time available to players when online lottery sales are not prohibited, while $R_t^{ban}$ represents the remaining leisure time after the reduction caused by increased $n_t$ following the ban on online lottery sales. The four diagonal lines in the figure correspond to the four equations mentioned above, as indicated in the upper right corner of the figure. It can be observed from the figure that $\Delta High$-lotto and $\Delta Low$-lotto exhibit little difference. This indicates that after the ban on online lottery sales, the increase in time cost for purchasing lottery tickets leads to a similar decline in lottery demand among high-income and low-income players. On the other hand, $\Delta High$-betting and $\Delta Low$-betting show significant differences, which can be attributed to the substantial variation in slopes. This suggests that the increase in time cost for purchasing lottery tickets after the ban disproportionately reduces the lottery demand among high-income players compared to low-income players. Consequently, the income structure of the betting-lottery becomes more skewed, with a larger proportion of low-income players.

In conclusion, an increase in the time needed to purchase lottery tickets, such as a rise in lottery purchase time costs due to the prohibition of lottery online sales, not only diminishes consumer demand for various lottery ticket types but also results in a more rapid decline in demand for betting-type lottery tickets among high-wage consumers compared to low-wage consumers. Drawing from the aforementioned theory, this study puts forth the following hypotheses:

 

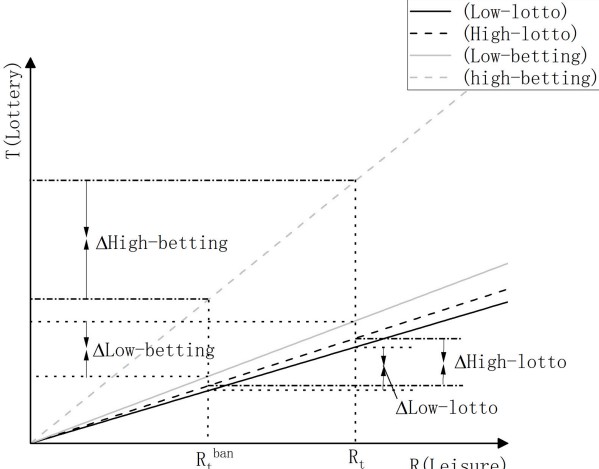

**Fig 1**. **A graph depicting the impact of income on the demand curve for lottery tickets.**

1. An increase in lottery purchase time costs will lead to a decrease in consumer demand for lottery tickets.
2. A rise in purchasing time costs will cause the rate of decline in demand for betting-lottery tickets among individuals in high-wage regions to outpace that of individuals in low-wage regions, while no significant disparity is observed for lotto-lottery tickets.

## Empirical research

### Econometric model

Following the implementation of China's policy to prohibit online lottery sales in March 2015, the overall growth rate of national lottery sales has markedly decelerated, even exhibiting negative growth. Subsequently, in October 2018, an additional policy concerning illegal lottery activities was enacted, further clamping down on internet sales, which consequently led to a precipitous drop in national lottery sales. Regarding the policy impact, this article adopts a general approach and applies the difference-in-differences (DID) method to conduct regression analysis on lottery and betting lottery sales in 2015 as the policy time node.

In the provincial panel data selected for this study, the total sales of sports lottery and welfare lottery each comprise sales data for four types of lottery tickets. The total sales of sports lottery encompass lotto-type, betting-type, instant-type, and video-type tickets. Since video-type sales account for less than 0.5% of the total, only the first three types of sales data are included in the analysis. The total sales of welfare lottery consist of lotto-type, instant-type, video-type, and keno-type tickets. Similarly, keno-type sales represent less than 0.5% of the total, so only the first three types of sales data are considered as well. In sports lottery total sales, lotto-type sales consistently account for over 50% of annual totals, which justifies their primary focus in the research.

The regulatory frameworks governing lotto-type play in sports lottery closely mirror those in welfare lottery. Against this background, this study initially employs sports lottery lotto-type sales to conduct regression-based correlation tests with welfare lottery lotto-type sales and sports lottery betting-type sales. These tests aim to examine potential associations between these sales volumes, requiring preliminary data correlation checks. Likewise, instant-type sales account for over 10% of both welfare and sports lottery totals, which warrants separate data correlation tests for these categories as well.

**The OLS model for data correlation test.** Given that the sales data of welfare lottery and sports lottery represent aggregates of multiple subtypes, the association between key subtypes, particularly lotto-type sales in both lottery systems and sports lottery betting-type sales, directly affects the suitability of using total sales data of welfare and sports lottery as dependent variables in subsequent analyses. In this section, classical OLS models are developed with sports lottery lotto-type, welfare lottery lotto-type, sports lottery instant-type, and welfare lottery instant-type sales as dependent variables, respectively. These models are used exclusively to identify associative patterns in the data rather than to infer causal relationships. This focus on association, rather than causal mechanisms, aligns with the analytical goal of capturing data-driven relationships between lottery subtypes.

$$\begin{cases} T_{it}^d = X_{1it}^d + X_{2t} + control_{1t} + control_{2t} + \varepsilon_{it}^d \\ ln T_{it}^d = ln X_{1it}^d + ln X_{2t} + ln control_{1t} + ln control_{2t} + \varepsilon_{it}^d \end{cases} \tag{15}$$

$$d = \begin{cases} 1, & \text{variable is of welfare lottery type} \\ 0, & \text{variable is of sports lottery type} \end{cases} \quad i = \begin{cases} 1, & \text{variable is of lotto-type} \\ 0, & \text{variable is of instant-type} \end{cases}$$

In the above models, $t$ is the time index. $T_{it}^d$ represents the dependent variable, where the superscript $d$ and the subscript $i$ are used to distinguish the four models. For example, when $d = 1$ and $i = 1$, it represents the welfare lottery lotto type; when $d = 1$ and $i = 0$, it represents the sports lottery lotto type; when $d = 0$ and $i = 1$, it represents the welfare lottery instant type; when $d = 0$ and $i = 0$, it represents the sports lottery instant type.

$X_{1it}^d$ represents the main explanatory variable, where the subscript 1 represents the other type of lottery in category $d$. For example, when $d = 0$, $X_{11t}^0$ represents the welfare lottery lotto type and $X_{12t}^0$ represents the welfare lottery instant type; when $d = 1$, $X_{11t}^1$ represents the sports lottery lotto type and $X_{12t}^1$ represents the sports lottery instant type. These are the same type of lottery data for different types of lotteries in the dependent variable. For example, when the dependent variable is the welfare lottery lotto type data, the main explanatory variable is the sports lottery lotto type data.

$X_{2t}$ represents the sales volume of sports lottery guessing type, which is the same variable in all four models.

$control_{1t}$ is a set of control variables, which consists of the remaining three types of welfare and sports lottery sales volumes excluding the dependent variable, the main explanatory variable, and the sports lottery guessing type.

$control_{2t}$ is a set of event control variables, including dummy control variables such as the World Cup, the European Cup, the ban on internet sales, and the COVID-19 pandemic.

$\varepsilon_{it}^d$ is a random disturbance term.

Using the method of Ordinary Least Squares (OLS) to estimate the correlation between the same types of sports and welfare lotteries, the linear unbiased estimator obtained is the best among all econometric methods. This is critical to whether the data used in subsequent empirical research in this paper is reasonable and effective.

**The DID model for the impact of banning online sales on welfare and sports lottery.** Based on the analysis of the previous theoretical model, in order to test the different impacts of banning online sales on the welfare and sports lotteries in high and low-income areas, this section subdivides the two types of lotteries, welfare and sports, and constructs a DID model with welfare lottery sales and sports lottery sales as dependent variables.

In our study, the DID model is adaptively adjusted: the "dummy variable distinguishing the treatment group from the control group" is replaced with a "dummy variable distinguishing high-income groups from low-income groups" (as shown in Model 16). However, the core calculation principle of double difference remains unchanged, and the specific logic is detailed in Table 1.

$$T_t^d = \beta_0 + \beta_1 banonline_t + \beta_2 high_i + \beta_3 banonline_t \cdot high_i + \beta_4 control_{it} + \varepsilon_{it} \tag{16}$$

**Table 1. The meanings of the parameters in model 16.**

| Welfare(Sports) lottery | Period 0 [1] (Banonline=0) | Period 1 [2] (Banonline=1) | Difference |
|---|---|---|---|
| high-income areas(high) | $\beta_0 + \beta_2 + \beta_4$ | $\beta_0 + \beta_1 + \beta_2 + \beta_3 + \beta_4$ | $\Delta T_1^d + \Delta T_2^d + \Delta T_3^d = \beta_1 + \beta_3$ |
| low-income areas(low) | $\beta_0 + \beta_4$ | $\beta_0 + \beta_1 + \beta_4$ | $\Delta T_1^d + \Delta T_2^d + \Delta T_3^d = \beta_1$ |
| DID | | | $\Delta\Delta T_1^d + \Delta\Delta T_2^d + \Delta\Delta T_3^d = \beta_3$ |

[1]the period before the ban on online sales.
[2]the period after the ban on online sales.

$$d = \begin{cases} w, \text{variable is the welfare lottery} \\ s, \text{variable is the sports lottery} \end{cases} \qquad T_t^d = \begin{cases} T_t^w = T_{1t}^w + T_{2t}^w + T_{3t}^w \\ T_t^s = T_{1t}^s + T_{2t}^s + T_{3t}^s \end{cases}$$

The coefficient $\beta_3$ of *Banonline · high* is of interest in this paper for the impact of banning online lottery sales on the rate of increase or decrease in lottery sales in high and low-income areas.

$control_{it}$ is a set of control variables, including per capita GDP, unemployment rate, housing prices, population size, average level of education, World Cup and COVID-19 pandemic among other factors.

$\varepsilon_{it}$ is a random disturbance term.

$T_t^d$ is the dependent variable, with the superscript $d$ used to distinguish between the two models where the dependent variable is either welfare or sports lottery. For example, when $d = w$, the dependent variable is the total sales of the welfare lottery; when $d = s$, the dependent variable is the total sales of the sports lottery.

According to the analysis in the theoretical model section, the most important subject of study in this section is the sales volume of sports lottery. The total sales volume of sports lottery $T^s$ is the sum of the sales volumes of three types: lotto $T_1^s$, instant $T_2^s$, and betting $T_3^s$; the total sales volume of welfare lottery $T^w$ is the sum of the sales volumes of three types: lotto $T_1^w$, instant $T_2^w$, and video $T_3^w$.

Using the DID method requires handling the treatment and control groups to satisfy the common trend assumption, which assumes that the growth trends of lottery sales in high-income and low-income areas before the national ban on online lottery sales do not exhibit systematic differences over time. Therefore, before conducting the DID analysis, it is necessary to perform a common trend test. Only when this test is passed can we conclude that the results of the DID analysis are reliable.

## Descriptive statistics

**Data source.** This study employs provincial panel data from 30 Chinese regions (2007-2021) to evaluate the differential impacts of internet lottery sales prohibition on sports and welfare lottery markets across income tiers, complemented by national time-series data (January 2012-December 2021) to analyze cross-system correlations in homologous lottery products (e.g., lottos vs. lottos). Post-2021 data exclusion derives from two unrecorded market responses: (a) Post-pandemic expansion of physical lottery outlets, with over 150,000 new stores licensed in 2022 (China Commerce Daily, 2023), substantially reduced consumers' time costs for offline purchases; (b) 83% of vendors adopted WeChat-mediated proxy betting during the 2022 FIFA World Cup (Tencent Consumer Report, 2023), enabling customers to commission purchases remotely via cash transactions that evaded official tracking. These undocumented adaptations would artificially attenuate the policy's observed effects by reintroducing time-efficient alternatives, necessitating temporal truncation for causal identification. Data were sourced from the China Statistical Yearbook, National Bureau of Statistics, People's Bank of China, CEID macroeconomic database, and lottery regulatory agencies. The Tibet Autonomous Region, Hong Kong, Macau, and Taiwan were excluded due to data availability constraints.

**Descriptive statistics for data correlation test.** This study aims to investigate the relationship between the sales of lotto and instant lottery tickets within the welfare and sports lottery sales data. To this end, the sales of lotto and instant lottery tickets within the welfare and sports lottery are selected as the dependent variables. The primary independent variable corresponds to the different types of lottery variables that match the dependent variable. For instance, when the welfare lottery data serves as the dependent variable, the sports lottery data is designated as the primary independent variable.

This study also incorporates control variables in the form of dummy variables, such as the prohibition of online sales, the occurrence of the World Cup, the European Cup, and the COVID-19 pandemic. Notably, the World Cup and European Cup variables are granular to the level of their respective start and end months.

Descriptive statistics for each variable discussed in this section are presented in the corresponding table (Table 2).

**Descriptive statistics for the DID regression test.** In the context of the Difference-in-Differences (DID) model delineated in Equation (16), the dependent variable is operationalized as the aggregate sales volume of both the sports and welfare lotteries across all provinces. The comprehensive sales volume of the sports lottery encapsulates the sales data from the lotto, instant, and quiz formats of the sports lottery. Similarly, the aggregate sales volume of the welfare lottery incorporates sales data from the lotto, instant, and video formats.

For the purpose of this study, provinces exhibiting a per capita GDP exceeding 30% of the national average GDP are classified into the high-income bracket, whereas provinces with a per capita GDP falling below 85% of the national average GDP are categorized into the low-income bracket. On a macroeconomic level, the model accounts for potential confounding factors including per capita GDP, unemployment rate, housing prices, population size, average educational attainment, the occurrence of the World Cup, and the impact of the COVID-19 pandemic.

Descriptive statistics pertaining to the variables discussed in this section are presented in Table 3.

**Table 2**. Descriptive statistics of the main variables in model (15).

| Variable | Meaning | Mean | Std. Dev. | Min | Max |
|---|---|---|---|---|---|
| W-lotto | welfare lottery lotto-type | 111.2451 | 26.22014 | 0 | 156.0731 |
| W-instant | welfare lottery instant-type | 14.09019 | 4.787208 | 0 | 35.83421 |
| W-video | welfare lottery video-type | 28.7439 | 11.94953 | 0 | 47.2341 |
| S-lotto | sports lottery lotto-type | 76.77431 | 18.41028 | 0 | 110.7114 |
| S-instant | sports lottery instant-type | 12.92999 | 4.247675 | 0.013 | 27.7334 |
| S-betting | sports lottery betting-type | 75.84142 | 46.83459 | 0 | 294.8631 |
| Banonline | ban on online lottery sales | 1.008333 | 0.8043561 | 0 | 2 |
| NCP | COVID-19 pandemic | 0.1208333 | 0.2592477 | 0 | 1 |
| Worldcup | FIFA World Cup | 0.0333333 | 0.1802581 | 0 | 1 |
| Eurocup | UEFA European Championship | 0.0416667 | 0.2006642 | 0 | 1 |

**Table 3**. Descriptive statistics of the main variables in model (16).

| Variable | Meaning | Mean | Std. Dev. | Min | Max |
|---|---|---|---|---|---|
| spotlottery | sports lottery total sales | 426661.3 | 451207.5 | 2410 | 2852390 |
| wellottery | welfare lottery total sales | 393712.4 | 320614.3 | 11116 | 1677928 |
| pergdp | per capita GDP | 3.309152 | 0.674539 | 1.21 | 4.57 |
| houseprice | housing prices | 7416.66 | 6618.876 | 1851 | 45000 |
| popula | population | 37.97553 | 26.0617 | 2.8883 | 99.4117 |
| peredu | per capita education | 8.856493 | 1.340715 | 4.221938 | 12.68113 |
| interest | benchmark interest rate | 4.549333 | 1.083525 | 3.6 | 7.65 |
| Banonline | ban on online lottery sales | 0.4666667 | 0.4996453 | 0 | 1 |
| Worldcup | FIFA World Cup | 0.2 | 0.4006074 | 0 | 1 |
| NCP | COVID-19 pandemic | 0.0866667 | 0.2556435 | 0 | 1 |

## Empirical results

The welfare lottery and sports lottery, serving as two prominent avenues for public welfare revenue within the jurisdiction of the Chinese Ministry of Finance, exhibit two overlapping game types among their offerings. Thus, before delving into the research scope of total sales for these lotteries, it is imperative to elucidate the intricate interconnections between the identical game types featured in both. By employing the venerable ordinary least squares (OLS) regression model to estimate the data pertaining to the corresponding game types, the most optimal linear unbiased estimators can be derived. A substantial correlation observed among the data associated with the similar game types across these two lotteries substantiates the reliability of the subsequent research findings presented herein.

**Results for the data correlation test.** Prior to conducting regression analysis using the total sales of welfare lottery ($Y^w$) and sports lottery ($Y^s$) as dependent variables, it is essential to examine the correlation between the primary components of the welfare lottery, namely, lottery-type ($Y_1^w$) and instant-win-type ($Y_2^w$), and the primary components of the sports lottery, namely, lottery-type ($Y_1^s$) and instant-win-type ($Y_2^s$). This analysis aims to investigate the relationship between the sales of similar lottery types between welfare and sports lotteries. In this study, the lottery-type ($S$–$lotto$) and instant-win-type ($S$–$instant$) of the sports lottery are considered as dependent variables, and eight models are formulated by considering controlled and uncontrolled variables, as well as the logarithmic and non-logarithmic scenarios. The regression results are presented in S1 Table.

In the regression results table (S1 Table), several models are presented to analyze the relationship between variables. Models (1) and (5) represent simple linear regression models without the inclusion of control variables or the application of logarithmic transformations. Models (2) and (3) extend the analysis by incorporating control variables while still excluding logarithmic transformations. Models (3) and (6) introduce a double logarithmic approach where both the explanatory and dependent variables are logarithmically transformed, albeit without the inclusion of control variables. Lastly, models (4) and (8) employ a double logarithmic framework, encompassing both logarithmic transformations and the inclusion of control variables.

In Model (4), the coefficient of determination $R^2$ is 0.902, and the $F$ statistic is 105.666, indicating a good fit and strong statistical association captured by the model. The results show that the correlation coefficient between welfare lottery lotto-type sales ($lnW$–$lotto$) and sports lottery lotto-type sales ($lnS$–$lotto$) is 0.956, which is significant at the 1% level. This indicates that a 1% change in welfare lottery lotto-type sales is associated with a 0.956% change in sports lottery lotto-type sales, reflecting a strong co-movement between the two variables in the data. The $R^2$ value of 0.902 further suggests that 90.2% of the variation in sports lottery lotto-type sales is statistically associated with variation in welfare lottery lotto-type sales, confirming a high degree of correlation.The results also show that, except for sports lottery betting-type sales ($lnS$-betting) in sports guessing games, all other lottery types exhibit a statistically significant association with sports lottery lotto-type sales.The correlation between sports guessing game sales and sports lottery lotto-type sales is less than 5%, indicating a weak associative pattern in this subset.

These findings are interpreted as evidence of statistical co-movement rather than causal relationships, as they reflect data-driven associations that may be influenced by shared underlying factors (e.g., seasonal trends, macroeconomic conditions) not disentangled in the current framework.

In Model (8), the coefficient of determination $R^2$ is 0.793, indicating that the model provides a moderate fit to the data. The $F$ statistic of 43.916 suggests that the model's explanatory power is statistically significant, although not exceptionally strong. The results demonstrate that the coefficient of the impact of welfare instant lottery sales ($lnW$–$instant$) on sports instant lottery sales ($lnS$–$instant$) is estimated to be 0.752, significant at the 1% level. This implies that a 1% change in welfare instant lottery sales corresponds to a 0.752% change in sports instant lottery sales. Hence, approximately 0.752% of the variation in sports instant lottery sales can be explained by changes in instant lottery sales, indicating a relatively high degree of correlation. It is worth noting that other lottery types do not exhibit a significant influence on sports instant lottery sales.

Furthermore, the remaining models in the regression results table (S1 Table) reveal a strong correlation in the sales variation of the same lottery type across different regions. Even when considering the national average results, the confidence intervals suggest that the correlation in the variation of the same lottery type across provinces does not exhibit substantial differences. These findings offer valuable insights for the subsequent analysis employing inter-provincial panel data in the context of difference-in-differences (DID) regression.

**Results for the common trend hypothesis test.** Prior to performing the Difference-in-Differences (DID) regression analysis, it is essential to examine the validity of the common trend hypothesis between the treatment group (high-income regions) and the control group (low-income regions). This test aims to determine whether the trends in welfare lottery and sports lottery sales were consistent between the high-income and low-income regions before the prohibition of cross-provincial lottery sales. Only when the common trend assumption holds true, can the results of the DID regression be deemed reliable. In this section, we employ both graphical and regression-based approaches to assess the presence of a common trend.

**Graphical Representation of Common Trends.** The graphical method is employed to visually examine the trends in welfare lottery and sports lottery sales for the high-income and low-income regions over time. S1 Fig and S2 Fig depict the sales trajectories, with the blue line representing sales in high-income regions and the red line representing sales in low-income regions. The horizontal axis represents the end of each year. Since the prohibition of internet lottery sales was implemented in March 2015, the time points in the figures are aligned with the end of each year, with the year 2014 corresponding to the depicted time point. Upon examination of both figures, it is evident that the trends before the 2014 time point were generally parallel, thus fulfilling the common trend assumption. S1 Fig demonstrates that even after the year 2014, the welfare lottery sales in high-income regions continue to exhibit a parallel trend with low-income regions. Similarly, in S1 Fig, during the period between 2014 and 2015, which corresponds to the first year of the internet sales prohibition policy, the decline in sports lottery sales in high-income regions is noticeably steeper compared to that in low-income regions.

**Regression Approach of Common Trends.** The regression-based method is employed to quantitatively assess the common trend hypothesis. S2 Table and S3 Table present the results of the common trend test for welfare lottery and sports lottery, respectively. Models (1) and (2) represent the fixed-effects model with time indicators, while models (3) and (4) represent the individual fixed-effects model with both time and individual indicators. In these tables, "high1," "high2," and "high3" represent the interaction terms between high-income regions and the years 2014, 2013, and 2012, respectively. To satisfy the common trend assumption, these three interaction terms should not be statistically significant, indicating that there were no significant differences in the growth rates of welfare lottery sales between high-income and low-income regions before the policy intervention.

The results presented in S2 Table indicate that in all four models, "high1," "high2," and "high3" are not statistically significant. This suggests that the regression results have passed the test of the common trend hypothesis, indicating that there were no significant differences in the growth rates of welfare lottery sales between high-income and low-income regions before the implementation of the internet sales prohibition policy.

The results in S3 Table indicate that in all four models, the coefficients of the variables "high1," "high2," and "high3" are not statistically significant. This implies that the regression results have passed the test of the common trend hypothesis, providing evidence that there were no significant differences in the growth rates of welfare lottery sales between high-income and low-income regions prior to the implementation of the internet sales prohibition policy.

Based on the comprehensive assessment of the common trend assumption using both graphical and regression methods, it can be concluded that the common trend assumption is upheld. As a result, the present study fulfills the prerequisites for conducting the DID regression analysis.

**Results for DID regression test.**  Based on the theoretical analysis presented earlier, the prohibition of online lottery sales is expected to have a negative impact on both welfare and sports lottery sales. However, the magnitude of this impact may vary across different income groups. Specifically, the decline in demand for welfare lottery sales may be similar among high-income and low-income individuals. Conversely, for sports lottery sales, high-income individuals may experience a more pronounced decrease in demand for sports betting compared to their low-income counterparts. In this section, we will perform separate DID regression analyses to examine the effects of the policy on welfare and sports lottery sales in high-income and low-income areas. The regression results are reported in Table 4.

In Table 4, Model (1) presents the DID regression results for welfare lottery sales following the implementation of the policy prohibiting online lottery sales. The estimated coefficient for the DID effect is -0.121, but it is not statistically significant at conventional levels. Model (2) builds upon Model (1) by incorporating individual and time fixed effects. The estimated DID coefficient in Model (2) is -0.095, which is also not statistically significant. These findings suggest that the implementation of the policy did not have a significant impact on the growth or decline rate of welfare lottery sales in high-income and low-income regions, aligning with the theoretical hypotheses of this study.

However, in the DID regression Model (3) for sports lottery, the estimated DID coefficient is -0.371, and it is statistically significant at the 1% level. Similarly, Model (4) includes individual and time fixed effects, and it reveals a statistically significant estimated DID coefficient of -0.426 at the 1% level. These results indicate that after the implementation of the policy, the rate of decline in sports lottery demand in high-income regions is approximately 37% to 42% higher compared to low-income regions. The prohibition of online lottery sales has led to a rapid decrease in demand for sports lottery in high-income regions, aligning with the theoretical expectations of this study.

According to Models (1) and (3), it is evident that large-scale sports events such as the World Cup have a significant positive influence on sports lottery sales. This effect can be attributed to the heightened enjoyment and engagement of individuals participating in lottery activities while following the World Cup. Conversely, no significant impact on welfare lottery sales was observed.

The global COVID-19 pandemic (referred to as NCP) has had a noteworthy adverse effect on lottery sales. This phenomenon can be attributed to the implementation of pandemic control measures, resulting in the closure of many physical lottery outlets and the suspension of online sales. Consequently, the increased transaction costs associated with purchasing lottery tickets and the limited accessibility have significantly diminished the demand for lottery products.

Furthermore, the benchmark interest rate (referred to as interest) exhibited a significant negative influence on lottery sales. This outcome is attributed to the positive relationship between the benchmark interest rate and the expected returns on a variety of financial assets. As a result, individuals' expectations for higher financial asset returns lead to a decreased demand for lottery products, as indicated by the Euler equation of lottery consumption (25).

It should be noted that the sports lottery sales data presented encompass the cumulative sales of lottery-style, instant-win, and guessing-type sports lottery products. Based on the previous correlation analysis, a strong correlation between welfare lottery-style and instant-win products and sports lottery-style and instant-win products was identified. Consequently, the policy prohibiting online sales did not significantly affect the sales of welfare lottery-style and instant-win products, implying a similar limited impact on the sales of sports lottery-style and instant-win products. Thus, in Models (3) and (4), the observed significant effects specifically pertain to the sales of sports lottery guessing-type products, aligning with the theoretical analysis conducted in the preceding sections of this study.

**Table 4**. DID regression results for welfare and sports lottery.

| | (1) lnwellottery | (2) lnwellottery | (3) lnspotlottery | (4) lnspotlottery |
|---|---|---|---|---|
| did | -0.121 | -0.095 | -0.371*** | -0.426*** |
| | (-1.31) | (-1.55) | (-3.40) | (-6.55) |
| Banonline | 0.184** | -0.339 | 0.494*** | 1.492*** |
| | (2.49) | (-1.50) | (5.69) | (6.21) |
| high | -0.095 | | 0.192 | |
| | (-0.92) | | (1.57) | |
| lnpergdp | 0.760*** | 0.554*** | 1.052*** | 0.133 |
| | (7.88) | (3.46) | (9.26) | (0.78) |
| lnumemploy | -0.200** | -0.569*** | 0.137 | -0.239* |
| | (-2.12) | (-4.71) | (1.23) | (-1.87) |
| lnpopula | 0.727*** | 1.193*** | 0.904*** | 1.557*** |
| | (28.87) | (3.70) | (30.45) | (4.55) |
| lnperedu | -0.464** | 1.708*** | -0.819*** | 1.053** |
| | (-2.52) | (4.10) | (-3.77) | (2.38) |
| Worldcup | 0.058 | | 0.145** | |
| | (1.04) | | (2.19) | |
| NCP | -0.543*** | | -0.253** | |
| | (-6.07) | | (-2.39) | |
| lninterest | -0.480*** | | -0.450*** | |
| | (-3.93) | | (-3.13) | |
| 2007.year | | 0.000 | | 0.000 |
| | | (.) | | (.) |
| 2008.year | | -0.096 | | 0.261*** |
| | | (-1.36) | | (3.49) |
| 2009.year | | 0.029 | | 0.373*** |
| | | (0.39) | | (4.70) |
| 2010.year | | 0.183** | | 0.506*** |
| | | (1.98) | | (5.14) |
| 2011.year | | 0.166 | | 0.642*** |
| | | (1.51) | | (5.50) |
| 2012.year | | 0.242** | | 0.776*** |
| | | (1.97) | | (5.94) |
| 2013.year | | 0.359*** | | 1.012*** |
| | | (2.66) | | (7.06) |
| 2014.year | | 0.502*** | | 1.326*** |
| | | (3.47) | | (8.63) |
| 2015.year | | 0.779*** | | -0.158 |
| | | (7.70) | | (-1.47) |
| 2016.year | | 0.761*** | | -0.088 |
| | | (8.15) | | (-0.89) |
| 2017.year | | 0.710*** | | -0.028 |
| | | (8.57) | | (-0.32) |
| 2018.year | | 0.667*** | | 0.254*** |
| | | (8.87) | | (3.18) |
| 2019.year | | 0.408*** | | -0.004 |
| | | (5.74) | | (-0.05) |
| 2020.year | | 0.140** | | -0.152** |
| | | (2.01) | | (-2.06) |
| 2021.year | | 0.000 | | 0.000 |
| | | (.) | | (.) |
| _cons | 2.944*** | -0.464 | -0.727 | 2.863 |
| | (3.48) | (-0.25) | (-0.73) | (1.46) |
| $R^2$ | 0.829 | 0.845 | 0.866 | 0.901 |
| adj. $R^2$ | 0.824 | 0.823 | 0.862 | 0.887 |
| F | 154.702 | 82.833 | 206.105 | 138.008 |

$t$ statistics in parentheses * $p < 0.1$, ** $p < 0.05$, *** $p < 0.01$

## Conclusion

Lottery public welfare funds play a crucial role as a significant source of government revenue. Enhancing lottery sales while simultaneously improving the structure of lottery participants and maximizing the impact of the third distribution are key objectives. With over 30 years of lottery development in China, various policies have been implemented to regulate the lottery market, but there are still several areas that require attention. An in-depth analysis of the impact of prohibiting online sales of lottery tickets is of paramount importance to lottery participants.

Firstly, this study employed the Bellman discrete-time dynamic optimization equation to derive the Euler equation of lottery consumption. The analysis revealed that consumers' expectations of real financial asset returns decrease in the face of negative news affecting financial asset yields, such as economic recessions, stock market crashes, and severe inflation, or when they lack access to financial asset returns due to limited capabilities. Consequently, such factors diminish consumers' expectations of financial asset returns and increase their demand for lottery tickets. By examining the relationship between consumers' marginal utility of leisure and the marginal utility of lottery tickets, it was found that an increase in the time cost associated with purchasing lottery tickets leads to a decrease in consumers' demand for such tickets. Moreover, individuals with more leisure time tend to exhibit a higher demand for lottery tickets. When comparing lotto-lottery tickets and betting-lottery tickets, it was observed that as the time cost of purchasing lottery tickets increases, the decline in demand for lotto-lottery tickets remains relatively consistent among high-income and low-income individuals. However, for betting-lottery tickets, the decline in demand is more pronounced among high-income individuals compared to their low-income counterparts.

Furthermore, this study employed a Difference-in-Differences (DID) empirical approach to evaluate the impact of the prohibition of online lottery ticket sales on the sales of welfare and sports lottery tickets in different income regions, based on panel data from 30 provinces in China spanning the period from 2007 to 2021. The findings revealed that the implementation of the policy to prohibit online sales of lottery tickets resulted in a faster decline in demand for sports betting-lottery tickets in high-income regions compared to low-income regions. However, there was no significant disparity in the decline of demand for welfare lotto-lottery tickets between high-income and low-income regions.

Moreover, the analysis based on theoretical hypothesis 1 demonstrated that events that reduce the convenience of purchasing lottery tickets, such as the prohibition of online sales of lottery tickets and the outbreak of the COVID-19 pandemic, had a significant negative impact on lottery sales.

From the perspective of balancing policy impacts and aligning with the public welfare nature of lotteries in China, the prohibition of online lottery purchases presents a double-edged sword effect on social health and public welfare goals. On the positive side, by reducing purchase convenience, it may effectively curb addictive lottery consumption, particularly protecting low-income groups from excessive spending, which aligns with the core mission of lotteries to serve public welfare and safeguard social well-being. On the negative side, it simultaneously hinders the widespread participation of high-income groups, whose rational, low-frequency, and small-amount lottery participation is critical for the sustainable growth of public welfare funds. Notably, the impact of such a ban is likely short-term, as convenience can be gradually restored through alternative channels like expanded physical lottery outlets, phone sales, or promotional efforts via short-video platforms.

Our study further reveals that policies aimed at enhancing purchase convenience and reducing time costs are particularly effective in driving sports lottery consumption among high-income individuals, which in turn helps optimize the demographic structure of lottery participants. However, it is important to clarify that this focus on convenience does not imply ignoring potential risks. We acknowledge that investigating whether the online ban has led to substitution toward other gambling products (such as foreign online platforms) requires more granular behavioral data, which is beyond the scope of the current research, and this remains a valuable direction for future inquiry.

Therefore, optimizing purchase convenience through prudent measures is crucial. In the context of potential future relaxation of online lottery systems, this necessitates reliance on an official unified platform for whole-process supervision, including the presetting of daily and weekly betting limits, the mandatory display of cumulative purchase duration and consumption amounts, and the triggering of automatic reminders or temporary restrictions for high-frequency and high-amount betting behaviors. This approach ensures convenience for ordinary users engaging in low-frequency purchases while directly mitigating addictive consumption risks through technical interventions. Such a framework of "official-led platforms integrated with technical rigid constraints" also holds international reference value, as it offers insights for other countries, particularly those with rapidly popularizing online sports lotteries and underdeveloped regulatory regimes, to balance sales growth and addiction prevention. By incorporating local income stratification characteristics, these countries can prioritize enhancing purchase thresholds through measures such as real-name authentication and account classification to screen for rational participants. Simultaneously, they can employ artificial intelligence algorithms to identify abnormal purchase patterns, such as high-frequency betting within short timeframes, for real-time intervention. This facilitates synergy between the sustainable accumulation of public welfare funds and the advancement of social health objectives, thereby better fulfilling the welfare-oriented mission of lotteries as encapsulated in the principle of "taken from the people, used for the people."

Lastly, it is essential to acknowledge certain limitations and areas for further research in this study. Due to data constraints related to specific core variables at the city level, provincial-level panel data were utilized as an alternative to investigate the impact of the prohibition of online lottery ticket sales. Consequently, the study could not conduct exhaustive tests or heterogeneous analyses of the theoretical results, providing only a preliminary insight into the overall landscape. Future research could delve deeper into the subject matter by exploring the effects of lottery prize caps on consumers' subjective expectations of returns, as well as the intertemporal substitution elasticity between consumers' subjective expectations of probability and lottery ticket consumption. Such endeavors would benefit from the continued evolution of the lottery industry, policy advancements, and the availability of comprehensive lottery platform data.

## Supporting information

**S1 Appendix. The derivation for value function.**
(PDF)

**S1 Fig. Graphical representation of welfare lottery common trends.**
(TIFF)

**S2 Fig. Graphical representation of sports lottery common trends.**
(TIFF)

**S1 Table. Regression results for correlation.**
(TIFF)

**S2 Table. Common trends test in welfare lottery.**
(TIFF)

**S3 Table. Common trends test in spots lottery.**
(TIFF)

## Author contributions

**Conceptualization:** Zeyu Feng.

**Data curation:** Zeyu Feng.

**Formal analysis:** Zeyu Feng.

**Methodology:** Zeyu Feng.

**Resources:** Zeyu Feng.

**Validation:** Zeyu Feng.

**Visualization:** Zeyu Feng.

**Writing – original draft:** Zeyu Feng.

**Writing – review & editing:** Zeyu Feng.

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
