## [Decision Letter · Decision Letter 0]

5 Jun 2025

PONE-D-25-16988Impact of Online Lottery Sales Prohibition on the Structure of Lottery Consumers: A Time Cost Perspective in ChinaPLOS ONE

Dear Dr. Feng,

Thank you for submitting your manuscript to PLOS ONE. After careful consideration, we feel that it has merit but does not fully meet PLOS ONE’s publication criteria as it currently stands. Therefore, we invite you to submit a revised version of the manuscript that addresses the points raised during the review process.

Both reviewers agree that the study is interesting but, with different extent, they suggest to expand the literature review and to consider in your paper - or at least to cite - also different types of lotteries, different frameworks, and to include related papers in the literature review. Both reviewers have a detailed list of concerns and I expect you'll be able to do the requested changes and to answer point by point to those lists. 

We look forward to receiving your revised manuscript.

Kind regards,

Fabio Rapallo, Ph.D.

Academic Editor

PLOS ONE

3. We note you have included a table to which you do not refer in the text of your manuscript. Please ensure that you refer to Table 1 in your text; if accepted, production will need this reference to link the reader to the Table.

Additional Editor Comments:

One of the reviewer's reports has been added as an attachment by me. Sorry for the incovenience.

Reviewers' comments:

Reviewer's Responses to Questions

**Comments to the Author**

1. Is the manuscript technically sound, and do the data support the conclusions?

Reviewer #1: Yes

2. Has the statistical analysis been performed appropriately and rigorously? 

Reviewer #1: Yes

3. Have the authors made all data underlying the findings in their manuscript fully available?

Reviewer #1: Yes

4. Is the manuscript presented in an intelligible fashion and written in standard English?

Reviewer #1: Yes

5. Review Comments to the Author

Reviewer #1: Feng provides a comprehensive assessment of the impact of online lottery prohibition on the development of China’s sports welfare lottery. The policy background is crucial: the initiative aims to combat illegal online betting, both domestic and international. Existing literature suggests that unrestricted sports betting, particularly among younger populations, negatively impacts personal development. Overall, this policy direction is significant, although it may have a secondary effect on sales. The study is thorough, and the methodology is appropriate, offering a balanced evaluation of the policy's impact.

However, there are a few areas that could be further refined. First, the discussion lacks a balanced interpretation of the results. While the policy may reduce sales, it could ultimately be a net positive for societal health. The lottery in China is intended to serve a welfare purpose, not a wealth distribution one. The author is encouraged to include this perspective in the discussion. Second, the practical implications, including international relevance, are not addressed. For example, as online lottery systems may be relaxed in the future, what recommendations would the author have to strike a balance between sales and addictive behaviors? Additionally, what implications does this have for other countries, especially as online sports lotteries gain popularity worldwide? Third, this is not a strictly DID analysis, as it lacks a control group. The author should acknowledge this limitation and consider other potential confounders, such as the increased sales during the World Cup in recent years.

6. PLOS authors have the option to publish the peer review history of their article (what does this mean?). If published, this will include your full peer review and any attached files.

Reviewer #1: No

---

## [Author Response · Author response to Decision Letter 1]

23 Jul 2025

Response to Reviewers

Reviewer #1

First, the discussion lacks a balanced interpretation of the results. While the policy may reduce sales, it could ultimately be a net positive for societal health. The lottery in China is intended to serve a welfare purpose, not a wealth distribution one. The author is encouraged to include this perspective in the discussion.

Revision Response:

While prohibiting online lottery purchases has significantly reduced overall lottery sales (particularly for sports lottery’s competitive betting products), there exists a structural disparity across different income groups: the decline in purchase volume is more pronounced in high-income regions. From the perspective of social health and the public welfare nature of lotteries in China, the policy’s impact presents a "double-edged sword" effect.

On the positive side: By reducing the convenience of purchasing, it may effectively curb some addictive lottery consumption (with a particularly prominent protective effect on low-income groups), which aligns with the core goal of lotteries to serve public welfare and safeguard public well-being.

On the negative side: It has also hindered the widespread participation of high-income groups, whose rational, low-frequency, and small-amount lottery participation is an important force supporting the sustainable growth of public welfare funds.

Therefore, optimizing the convenience of lottery purchases (such as cautiously regulating online channels) is crucial for improving the structure of lottery participants. It can not only expand the participation of high-income groups but also reduce addiction risks through technical controls (e.g., daily betting limits, purchase duration warnings). Ultimately, this will promote the synergy between public welfare fund accumulation and social health development, better fulfilling the welfare mission of lotteries as "taken from the people, used for the people."

Reviewer #1:

Second, the practical implications, including international relevance, are not addressed. For example, as online lottery systems may be relaxed in the future, what recommendations would the author have to strike a balance between sales and addictive behaviors? Additionally, what implications does this have for other countries, especially as online sports lotteries gain popularity worldwide?

Revision Response:

In response to the reviewer’s concern regarding the lack of practical implications and international relevance, the following targeted revisions have been made to the conclusion section:

Recommendations for balancing sales and addictive behaviors amid potential future relaxation of online lottery systems: A specific regulatory framework has been added, emphasizing "reliance on an official unified platform for whole-process supervision, including the presetting of daily and weekly betting limits, mandatory display of cumulative purchase duration and consumption amounts, and triggering of automatic reminders or temporary restrictions for high-frequency and high-amount betting behaviors." This measure ensures convenience for ordinary users engaging in low-frequency purchases while directly mitigating addictive consumption risks through technical interventions, addressing the concern of balancing sales growth and addiction prevention.

Supplementary analysis of international relevance: The framework of "official-led platforms integrated with technical rigid constraints" is explicitly highlighted as having international reference value. It provides insights for countries with rapidly popularizing online sports lotteries and underdeveloped regulatory regimes: by incorporating local income stratification characteristics, these countries can prioritize enhancing purchase thresholds through measures such as real-name authentication and account classification to screen for rational participants. Simultaneously, artificial intelligence algorithms can be employed to identify abnormal patterns (e.g., high-frequency betting within short timeframes) for real-time intervention, ultimately facilitating synergy between the sustainable accumulation of public welfare funds and the advancement of social health objectives.

These revisions strengthen the study’s practical implications and international relevance through concrete policy recommendations and cross-country adaptability analysis.

Reviewer #1:

Third, this is not a strictly DID analysis, as it lacks a control group. The author should acknowledge this limitation and consider other potential confounders, such as the increased sales during the World Cup in recent years.

Revision Response:

We appreciate Reviewer #1’s third comment. Regarding the concern that "this is not a strictly DID analysis due to the lack of a control group," we acknowledge that the design of DID in this study differs from the traditional DID in terms of control group setting, but the core analytical principle remains consistent.

The traditional DID model includes three key variables: a dummy variable distinguishing the period before and after a policy/intervention (Postₜ in Figure 1), a dummy variable distinguishing the treatment group from the control group (Treated in Figure 1), and their interaction term (Post×Treated). Its calculation logic is further illustrated in Table 2: within the traditional framework, the differences in outcomes (ΔY₁, ΔY₀) between the treatment group (i=1) and control group (i=0) before (Period 0, Postₜ=0) and after (Period 1, Postₜ=1) the policy implementation are used to calculate the double difference effect (ΔΔY=β₃), i.e., the net effect of the policy.

Fig 1 Traditional DID Analytical Model

Table 2. Interpretation of Parameter Meanings in the Traditional DID Model

Y Period 0

(Postt=0) Period 1

(Postt=1) Difference

Treatment grop(i=1) Y10=α+β2 Y11=α+β1+β2+β3 ΔY1=β1+β3

Control grop(i=0) Y00=α Y01=α+β1 ΔY0=β1

DID ΔΔY=β3

In our study, the DID model is adaptively adjusted: the "dummy variable distinguishing the treatment group from the control group" is replaced with a "dummy variable distinguishing high-income groups from low-income groups" (as shown in Model 16). However, the core calculation principle of double difference remains unchanged, and the specific logic is detailed in Table 1.

Regarding the "other potential confounders, such as the increased sales during the World Cup in recent years" mentioned by the reviewer, these factors have been incorporated into the model through control variables (controlᵢₜ). controlᵢₜ includes a series of exogenous variables that may affect the outcomes, specifically: FIFA World Cup, COVID-19 pandemic, benchmark interest rate, population, per capita GDP, per capita education, and housing prices.

Reviewer #2:

In the introduction, the author makes statements that are highly improbable and may be inaccurate. For instance, it is stated that “The earliest research on lotteries traces back to Von Neumann and Mogenstern (1944), who proposed the expected utility theory.” Modern style lotteries have been available as far as the 15th century, in Europe, and became more prevalent in the 18th century. In that sense, it is incorrect to mention that the “earliest research on lotteries traces back” to 1944, as many early scholars have approached the topic albeit in different areas, but the author does not address that.

Revision Response:

We appreciate Reviewer #2’s insightful comment on the inaccuracy in the introduction. Regarding the statement that "the earliest research on lotteries traces back to Von Neumann and Mogenstern (1944), who proposed the expected utility theory," we acknowledge its imprecision and would like to clarify and revise it.

As noted by the reviewer, modern-style lotteries emerged as early as the 15th century in Europe and became more prevalent in the 18th century. Our original wording failed to distinguish between "the history of lottery practice" and "academic research on lotteries," leading to ambiguity. In fact, what we intended to emphasize is that within the framework of modern economic theory, Von Neumann and Mogenstern’s (1944) expected utility theory laid a foundational groundwork for systematic academic research on lottery-related behaviors. This theory provided core analytical tools for subsequent studies on risk preference, utility evaluation, and decision-making in lottery participation.

To address the oversight of early scholars’ discussions on lottery-related topics in other fields, we will supplement relevant historical context in the revised manuscript, clearly differentiating between the long history of lottery practice and the starting point of modern academic research on this subject to avoid misunderstanding.

Reviewer #2:

The manuscript broadens the scope from theoretical models to empirical, demographic-driven research, recognizing heterogeneous motivations across populations. However, the author fails to provide appropriate support on the relevance of the study and the research gap. Additionally, although the review includes literature up to 2024, which is commendable, it would benefit from more engagement with online/mobile lottery behavior, especially given the changing landscape of gambling.

Revision Response:

We sincerely appreciate Reviewer #2’s valuable feedback, which has helped strengthen the manuscript. We have addressed the concerns raised by enhancing the relevance of the study, clarifying research gaps, and deepening engagement with online/mobile lottery behavior literature, as detailed below:

1. Strengthening Support for Study Relevance and Research Gaps

To address the need for clearer support on relevance and research gaps, we have revised the literature review to explicitly identify critical limitations in existing scholarship within the context of digitalized gambling:

Under-exploration of purchasing convenience across demographics and lottery types: While prior research acknowledges purchasing convenience as a factor in consumption, it rarely examines how this factor shapes lottery behavior differentially across income groups or lottery categories (e.g., fixed-prize vs. progressive jackpots). This is a key gap, as offline studies have already documented income-based disparities in lottery participation, but their digital extensions remain underexplored.

Mismatch between traditional theories and digital contexts: Classical theoretical frameworks, developed for offline settings, struggle to explain the unique dynamics of digital lottery environments—such as real-time accessibility and mobile interactions—that reshape consumption decisions.

Overemphasis on sports betting in online gambling research: Existing online gambling studies focus heavily on sports betting, which differs from lotteries in core motives (skill vs. luck) and audience structure, leaving critical gaps in understanding online lottery-specific mechanisms.

These gaps underscore the relevance of our study: by focusing on temporal costs and leveraging China’s unique policy context of "prohibiting online lottery sales" as a natural instrument, we address these limitations to provide new insights into digital lottery behavior.

2.Enhancing Engagement with Online/Mobile Lottery Behavior Literature

We have significantly expanded engagement with online/mobile lottery behavior research, integrating key findings that reflect the changing gambling landscape:

Psychological and structural mechanisms: Early studies highlight how mobile platforms, enabled by smartphone accessibility, interact with associative learning to accelerate maladaptive behaviors, while digitized lottery markets drive addiction through omnilocal accessibility and transactional immediacy—factors that eliminate physical constraints, truncate deliberation intervals, and amplify impulsivity (James et al., 2017; Churchill and Farrell, 2018; Zhang et al., 2022).

Behavioral drivers and impacts: Subsequent research clarifies that real-time excitement, smartphone-enabled features (e.g., "cash-out"), and targeted marketing shape online gambling engagement, particularly among young adults, with promotional tactics reducing perceived risk and fueling impulsive behavior (Killick and Griffiths, 2021; Dunlop and Ballantyne, 2021; Killick and Griffiths, 2022; Reeve and Pincin, 2025).

Digital efficiency and expenditure: Recent work underscores that instantaneous, ubiquitous access to betting apps—coupled with platform functionality and social influences—facilitates harmful behaviors (e.g., increased frequency, impulsive wagering) among young adults, while U.S. states with high online lottery penetration demonstrated faster post-pandemic revenue recovery, confirming temporal efficiency as a critical driver of amplified spending (Hing et al., 2024; Hickman, 2025).

These additions ensure the review comprehensively reflects the multifaceted impact of digitalization on gambling behavior, aligned with the evolving landscape of online/mobile lottery.

We believe these revisions effectively address the reviewer’s concerns, enhancing the manuscript’s rigor and relevance. Thank you again for your constructive input.

Reviewer #2:

Also worth noting that while the study cites much research, the review lacks critical

engagement or evaluation of the limitations of the theories it cites. Are there empirical shortcomings in applying MEU or prospect theory to real-world lottery behavior? How do digital lotteries or gamification challenge classical models?

Revision Response:

We appreciate Reviewer #2’s insightful observation regarding the need for critical engagement with the theoretical limitations cited in the manuscript. To address this, we have incorporated an explicit evaluation of the empirical shortcomings of prospect theory and Maxmin Expected Utility (MEU) theory in explaining real-world lottery behavior, particularly in digital contexts.

Specifically, we note that prospect theory, with its focus on static decision-making and challenges in accounting for individual parameter variability, fails to capture dynamic adjustments like loss chasing and social influences on lottery participation. Meanwhile, MEU theory’s assumptions of perfect rationality and fixed risk attitudes cannot explain the overestimation of small probabilities, entertainment-driven motives, or context-dependent risk preferences inherent in lottery behavior. These limitations are further amplified by digital lotteries, where gamified features like instant feedback distort risk perceptions in ways that classical models cannot encompass.

This critical engagement with theoretical limitations strengthens the manuscript’s rigor and contextualizes the need for our study, which leverages a temporal cost framework to address these gaps in understanding digital lottery behavior. Thank you for highlighting this important aspect.

Reviewer #2:

The text can also benefit from more clarity and an improved writing style. Several phrases are overly complex or awkwardly constructed. As an example: “demonstrating that utility functions with quasi-concave and convex segments could rationalize the welfare-enhancing role of gambling...”. The author should also be cautious with using repetitive terms, such as “non-wealth motivations,” which appear multiple times without further nuance or variation.

Revision Response:

We sincerely appreciate Reviewer #2’s feedback on enhancing clarity, refining writing style, and addressing repetitive terminology. To address these concerns, we have made the following revisions:

Removing overly complex phrasing: The sentence referencing Hartley and Farrell (2002), which was noted as awkwardly constructed, has been removed from the manuscript. This streamlines the narrative without compromising the core theoretical framework.

Adding nuance to repetitive terms: For the term “non-wealth motivations,” we have added a concise explanation after its first appearance: “i.e., non-monetary benefits such as entertainment, social interaction, or the psychological thrill of anticipation”. This clarifies the concept and reduces redundancy in its subsequent mention, providing greater nuance to the discussion.

These revisions aim to improve readability and precision, ensuring the literature review is more accessible while main

---

## [Decision Letter · Decision Letter 1]

24 Sep 2025

PONE-D-25-16988R1Impact of Online Lottery Sales Prohibition on the Structure of Lottery Consumers: A Time Cost Perspective in ChinaPLOS ONE

Dear Dr. Feng,

Thank you for submitting your manuscript to PLOS ONE. After careful consideration, we feel that it has merit but does not fully meet PLOS ONE’s publication criteria as it currently stands. Therefore, we invite you to submit a revised version of the manuscript that addresses the points raised during the review process.

We look forward to receiving your revised manuscript.

Kind regards,

Madhabendra Sinha, PhD in Economics

Academic Editor

PLOS ONE

Journal Requirements:

Additional Editor Comments:

Reviewer #2: Minor Revision

Please revise accordingly and resubmit.

Reviewers' comments:

Reviewer's Responses to Questions

**Comments to the Author**

1. If the authors have adequately addressed your comments raised in a previous round of review and you feel that this manuscript is now acceptable for publication, you may indicate that here to bypass the “Comments to the Author” section, enter your conflict of interest statement in the “Confidential to Editor” section, and submit your "Accept" recommendation.

Reviewer #1: All comments have been addressed

Reviewer #2: All comments have been addressed

2. Is the manuscript technically sound, and do the data support the conclusions?

Reviewer #1: Yes

Reviewer #2: Partly

3. Has the statistical analysis been performed appropriately and rigorously? 

Reviewer #1: Yes

Reviewer #2: Yes

4. Have the authors made all data underlying the findings in their manuscript fully available?

Reviewer #1: Yes

Reviewer #2: Yes

5. Is the manuscript presented in an intelligible fashion and written in standard English?

Reviewer #1: Yes

Reviewer #2: Yes

6. Review Comments to the Author

Reviewer #1: All comments have been addressed by the authors. The paper can be accepted as is. (Minimal words count requirement: All comments have been addressed by the authors. The paper can be accepted as is.)

Reviewer #2: All the review comments have been properly addressed. However, some changes are still required for the sake of clarity.

A brief statement regarding the different types of lottery options available in China should be mentioned in the introduction section. Sports lottery and lotto lottery are the only options? Do the consumers have the options to purchase traditional type lottery tickets? Unless a brief introduction regarding the lottery system (as is prevalent in China) is discussed it is very difficult to judge the models developed by the author. It is really needed to know, whether the consumers are required to bear other costs (like brokerage fees etc.). Are all types of lotteries are being sold online? These questions are extremely essential to provide a proper judgement regarding the assumptions, econometric works and conclusions of the manuscript. Actually a brief introduction regarding the structure of lottery system in china is needed to be discussed.

1)The statement’ Does the reduction in lottery purchasing accessibility disproportionately affect high-income versus low-income groups?’ is grammatically correct but it can be improved using any of the following options

a) "Does the reduction in lottery purchasing accessibility disproportionately affect high-income or low-income groups?" or

b) "Does the reduction in lottery purchasing accessibility disproportionately impact high-income groups compared to low-income groups?"

2) ‘China’s per capita expenditure ($49) exhibits marked regional heterogeneity, lagging behind high-income nations (Norway: $864; Switzerland: $632) and even trailing some middle-income economies?’ Does it mean annual consumption expenditure per capita or per capita expenditure on lottery? Clarify properly. The subsequent statement may lead one to think that it is per capita consumption on lottery, however it should to clarified properly to have an idea with respect to the later part of the statement.

7. PLOS authors have the option to publish the peer review history of their article (what does this mean?). If published, this will include your full peer review and any attached files.

Reviewer #1: **Yes: **Yang Zhang

Reviewer #2: **Yes: **Dr. Sreemanta Sarkar

---

## [Author Response · Author response to Decision Letter 2]

8 Oct 2025

Response to Reviewers

Reviewer #2:

A brief statement regarding the different types of lottery options available in China should be mentioned in the introduction section. Sports lottery and lotto lottery are the only options? Do the consumers have the options to purchase traditional type lottery tickets? Unless a brief introduction regarding the lottery system (as is prevalent in China) is discussed it is very difficult to judge the models developed by the author. It is really needed to know, whether the consumers are required to bear other costs (like brokerage fees etc.) Are all types of lotteries are being sold online? These questions are extremely essential to provide a proper judgement regarding the assumptions, econometric works and conclusions of the manuscript. Actually a brief introduction regarding the structure of lottery system in china is needed to be discussed.

Revision Response:

We sincerely appreciate Reviewer #2’s valuable feedback on supplementing the introduction of China’s lottery system, which is critical for evaluating the manuscript’s models, assumptions, and conclusions. To address these concerns, we have added a brief statement in the introduction section, clarifying key details of China’s lottery system as follows:

China’s lottery market operates under a state monopoly, exclusively administered by two entities: the China Sports Lottery (CSL) and the China Welfare Lottery (CWL). The CSL offers three main types of lottery products: numerical lotteries (e.g., Super Lotto), sports-betting games (e.g., football match result lotteries), and instant scratch cards. The CWL primarily provides traditional draw games (e.g., Double Color Ball) and instant products. In terms of distribution, online lottery sales have been banned since 2015; currently, over 95% of transactions rely on more than 200,000 physical outlets nationwide. Consumers only pay the face value of lottery tickets (e.g., ¥2 per bet for Double Color Ball) with no additional costs such as brokerage fees.

Reviewer #2:

1)The statement’ Does the reduction in lottery purchasing accessibility disproportionately affect high-income versus low-income groups?’ is grammatically correct but it can be improved using any of the following options

a) "Does the reduction in lottery purchasing accessibility disproportionately affect high-income or low-income groups?" or

b) "Does the reduction in lottery purchasing accessibility disproportionately impact high-income groups compared to low-income groups?"

Revision Response:

We thank Reviewer #2 for pointing out the room for improving the sentence’s expression. Following the suggested option (a), we have revised the sentence in the manuscript to: “Does the reduction in lottery purchasing accessibility disproportionately affect high-income or low-income groups?”

Reviewer #2:

‘China’s per capita expenditure ($49) exhibits marked regional heterogeneity, lagging behind high-income nations (Norway: $864; Switzerland: $632) and even trailing some middle-income economies?’ Does it mean annual consumption expenditure per capita or per capita expenditure on lottery? Clarify properly. The subsequent statement may lead one to think that it is per capita consumption on lottery, however it should to clarified properly to have an idea with respect to the later part of the statement.

Revision Response:

We appreciate Reviewer #2’s reminder to clarify the definition of “per capita expenditure,” which helps avoid ambiguity. We have revised the relevant content in the manuscript to explicitly specify the indicator, as follows:

China's annual per capita consumption expenditure on lottery products ($49) exhibits marked regional heterogeneity. This figure significantly lags behind that of high-income nations (e.g., Norway: $864; Switzerland: $632) and even some middle-income economies in terms of annual per capita lottery expenditure.

---

## [Decision Letter · Decision Letter 2]

9 Nov 2025

Impact of Online Lottery Sales Prohibition on the Structure of Lottery Consumers: A Time Cost Perspective in China

PONE-D-25-16988R2

Dear Dr. Feng,

We’re pleased to inform you that your manuscript has been judged scientifically suitable for publication and will be formally accepted for publication once it meets all outstanding technical requirements.

Kind regards,

Madhabendra Sinha, PhD in Economics

Academic Editor

PLOS ONE

Additional Editor Comments (optional):

The revised version nicely addressed all comments.

Reviewers' comments:

Reviewer's Responses to Questions

**Comments to the Author**

1. If the authors have adequately addressed your comments raised in a previous round of review and you feel that this manuscript is now acceptable for publication, you may indicate that here to bypass the “Comments to the Author” section, enter your conflict of interest statement in the “Confidential to Editor” section, and submit your "Accept" recommendation.

Reviewer #1: All comments have been addressed

Reviewer #2: All comments have been addressed

2. Is the manuscript technically sound, and do the data support the conclusions?

Reviewer #1: Yes

Reviewer #2: Yes

3. Has the statistical analysis been performed appropriately and rigorously? 

Reviewer #1: Yes

Reviewer #2: Yes

4. Have the authors made all data underlying the findings in their manuscript fully available?

Reviewer #1: Yes

Reviewer #2: Yes

5. Is the manuscript presented in an intelligible fashion and written in standard English?

Reviewer #1: Yes

Reviewer #2: Yes

6. Review Comments to the Author

Reviewer #1: All comments from this reviewer have been addressed. All comments from this reviewer have been addressed.

Reviewer #2: The author has addressed all the issues in the revised version of the manuscript. This manuscript may be accepted.

7. PLOS authors have the option to publish the peer review history of their article (what does this mean?). If published, this will include your full peer review and any attached files.

Reviewer #1: **Yes: **Yang Zhang

Reviewer #2: **Yes: **Dr. Sreemanta Sarkar

---

## [Editor Report · Acceptance letter]

PONE-D-25-16988R2

PLOS ONE

Dear Dr. Feng,

I'm pleased to inform you that your manuscript has been deemed suitable for publication in PLOS ONE. Congratulations! Your manuscript is now being handed over to our production team.

Kind regards,

on behalf of

Dr. Madhabendra Sinha

Academic Editor

PLOS ONE